# Association between Dietary Habit and Clinical Parameters in Patients with Chronic Periodontitis Undergoing Supportive Periodontal Therapy

**DOI:** 10.3390/nu14234993

**Published:** 2022-11-24

**Authors:** Shinichi Tabe, Yohei Nakayama, Ryoki Kobayashi, Kstsunori Oyama, Daisuke Kitano, Jun Ogihara, Hidenobu Senpuku, Yorimasa Ogata

**Affiliations:** 1Department of Periodontology, Nihon University School of Dentistry at Matsudo, 2-870-1 Sakaecho-nishi, Matsudo 271-8587, Japan; 2Research Institute of Oral Science, Nihon University School of Dentistry at Matsudo, 2-870-1 Sakaecho-nishi, Matsudo 271-8587, Japan; 3Department of Microbiology and Immunology, Nihon University School of Dentistry at Matsudo, 2-870-1 Sakaecho-nishi, Matsudo 71-8587, Japan; 4Department of Computer Science, College of Engineering, Nihon University, 1 Tamuramachi, Tokusada, Nakagawara, Koriyama 963-8642, Japan; 5Division of Cardiology, Department of Medicine, Nihon University School of Medicine, 30-1 Oyaguchikamicho, Itabashi-ku, Tokyo 173-8610, Japan; 6Laboratory of Applied Microbiology and Biotechnology, College of Bioresource Sciences, Nihon University, 1866 Kameino, Fujisawa 252-8510, Japan

**Keywords:** periodontitis, nutritional factors, supportive periodontal therapy, multiple linear regression

## Abstract

The recurrence risk evaluation has been emphasized in periodontal stabilization during supportive periodontal therapy (SPT). However, nutritional factors, e.g., dietary habits such as the frequency of eating vegetables, are rarely included in the evaluation. In this study, the effect of nutritional factors on clinical periodontal parameters was examined in a lifestyle-related investigation and a periodontal examination in patients with periodontitis undergoing SPT. A total of 106 patients were recruited. Tendencies toward a negative correlation were found between rate of a probing depth (PD) of 4–5 mm, rate of PD ≥ 6 mm, the bleeding on probing (BOP) rate, periodontal inflamed surface area (PISA), and various nutritional factors. The number of teeth was a clinical parameter with a significantly high R2 (≥0.10) influenced by environmental factors, whereas PD, PD of 4–5 mm, the BOP rate, and PISA were influenced by nutritional factors. These results suggested that environmental factors reflected clinical parameters showing long-term pathophysiology, such as the PD rate. Nutritional factors tended to affect the current inflammatory pathophysiology, such as the BOP rate, PISA, and PISA/periodontal epithelial surface area. Therefore, environmental and nutritional factors appear to be useful for evaluating the risk of periodontitis during SPT.

## 1. Introduction

Oral diseases, including periodontitis, seriously affect systemic health and the quality of life (QoL) in a large number of individuals, and they may affect various aspects of life [1], such as diabetes [2], rheumatoid arthritis [3], and pregnancy [4]. Periodontitis is the most common cause of tooth loss in oral diseases, which may affect dietary habits due to chewing ability [5]. Chronic periodontitis is dependent on periodontopathic bacteria and the host immune response and progresses with inflammatory changes [6,7]. As periodontitis sometimes recurs after treatment of a periodontal lesion, re-evaluation is considered important to realize slight changes in periodontal condition. Age, smoking status (current smoker, former smoker, or nonsmoker), the Brinkman index (Br index) [8], and body mass index (BMI) [9] are well-known environmental factors of periodontitis. The case definitions for periodontitis were developed in the context of the 2017 World Workshop on the Classification of Periodontal and Peri-Implant Diseases and Conditions, and the items used to evaluate the progression of chronic periodontitis are graded [10]. Among the evaluation methods, smoking status, presence of diabetes mellitus, hemoglobin A1c (HbA1c) values, and high sensitivity C-reactive protein (hsCRP) in whole blood are contained as environmental and systemic risk factors. A close link has been found between inflammatory diseases and nutritional health, and periodontal disease has been recognized as a lifestyle-related disease. However, no evaluations have been conducted regarding nutritional status, including vitamin A [11] and calcium [12] levels, and environmental factors such as obesity [13].

The evaluation of risk factors for the periodontal condition during supportive periodontal therapy (SPT) was developed in 2003 [14]. The evaluation items include the number of teeth with a probing depth (PD) > 5 mm, the bleeding on probing (BOP) rate, the ratio between age and bone resorption, the number of missing teeth, systemic disease, and smoking status. Each item is evaluated on a three-point scale, and comprehensive risk is determined by the number of risk items. However, nutritional status is not included in the evaluation.

The relationship between multiple nutritional factors and the periodontal condition has been investigated, with most studies reporting the attenuation of periodontal resistance to pathogens resulting from nutritional imbalances, such as a lack of vitamin C [15], vitamin E [12], or carotenoids. The results in those studies were attributed to antioxidant effects and the association of calcium and magnesium with bone metabolism [16]. However, these etiological studies have utilized a considerable number of patients with/without periodontitis who underwent periodontal examination with partially measured sites. Moreover, most of the investigated clinical parameters were PD and clinical attachment level, and no items were used to evaluate inflammatory activity, such as the BOP rate and the periodontal inflamed surface area (PISA) [17].

Several questionnaires with oral symptoms have been widely used to investigate the validation of oral health, such as the child oral health impact profile (COHIP) [18]. Lower COHIP scores were significantly associated with the self-perception of poor general or oral health. The relationship between smoking, sugar-sweetened beverage (SSB) consumption, and tooth brushing among adolescents was investigated using questionnaires in China, demonstrating that adolescent smokers had a higher level of soft drink consumption, worse oral hygiene habits, and poorer oral health-related quality of life (OHRQoL) [19]. The differences in QoL between children and adolescents with chronic rhinitis and sinusitis and healthy children and adolescents were evaluated by questionnaire, and it was clarified that the greatest impairment to well-being in children with chronic rhinitis and sinusitis was the impact of the child’s health status on parents’ emotions, pain and discomfort, and general perception of health [20]. These reports suggested that a questionnaire is one of the most effective methods to collect a lot of items from patients before statistical analysis.

In this study, to examine whether nutritional factors can be utilized in risk assessments for periodontal disease, we statistically analyzed the relationship between the results of a questionnaire about dietary habits and clinical parameters during SPT. Moreover, we clarified the clinical parameters reflecting nutritional factors. The determined clinical parameters may be useful as a risk evaluation item for periodontitis promoted by nutrition factors in the SPT stage.

## 2. Materials and Methods

### 2.1. Subjects

Patients who were undergoing periodontal therapies and had been continuing SPT for more than two years were recruited for this study. Among patients who were undergoing blood tests in the hospital at the Nihon University School of Dentistry at Matsudo, we selected those without systemic diseases that are risk factors for periodontal disease, such as diabetes mellitus, regardless of age, sex, and prescription medicines. Written informed consent was obtained from all patients after they were given a full explanation of the study contents and methods. The exclusion criteria were incomplete periodontal examinations, acute oral cavity inflammations, and abscesses around the teeth.

### 2.2. Periodontal Examinations

Periodontal examinations were performed to record tooth mobility, PD, BOP [21], and plaque control records [22]. The measurements were performed as previously described [23]. The lifestyle questionnaire was administered during the patients’ most recent session of periodontal therapy (SPT2). To confirm improvements in clinical parameters from periodontal therapy, data from periodontal examinations at the beginning (baseline (BL)), re-evaluation (RE), and start of SPT (SPT1) were collected from electronic clinical records. The data from a total of four treatment stages (BL, RE, SPT1, and SPT2) were used to calculate the periodontal epithelial surface area (PESA) [17], PISA, PISA/PESA, and number of missing molars. The Eichner classification was converted into numerals from the record as described in a previous study [24].

### 2.3. Lifestyle Questionnaire

The lifestyle habits of each patient were obtained at SPT2. The questionnaire included items on nutritional factors, environmental factors, sex, age, medical history, current systemic diseases, prescription medicines, height, and weight. The environmental factors of periodontitis included smoking status (current smoker, former smoker, or nonsmoker), the Br index, and BMI, which were recorded and calculated from the questionnaire items (Appendix A). The nutritional questionnaire items were selected and partially modified based on the 2017 National Health and Nutrition Survey (Physical Situation Questionnaire) on dietary habits (Appendix A). The questionnaire responses were obtained from self-reports.

### 2.4. Analysis of the Relationship between Clinical Parameters and Environmental and Nutritional Factors

The environmental factors (smoking, Br index, and BMI) associated with clinical parameters (11 items) and nutritional factors (23 items) were statistically analyzed. Spearman’s rank correlation coefficients between the clinical parameters and environmental (four items) and nutritional factors (ten items) were calculated (Figure 1). For the multiple regression analysis, the clinical parameters that showed high correlation coefficients with environmental and nutritional factors were used as objective variables (eight variables in total), and four environmental and eight nutritional factors that showed high correlation coefficients with objective variables were used as explanatory variables.

### 2.5. Quantitative Polymerase Chain Reaction (PCR)

To investigate inflammatory status, total RNA was extracted from collected saliva using the Total RNA Life Plus Kit (NORGEN BIOTEK CORP, 3430 Schmon Parkway, Thorold, ON, Canada). Approximately 2.0 mL of non-stimulated saliva was collected from each patient in specialized containers (SALIVA COLLECTION AID, Cryogenic Vials, SalivaBio) at SPT2 and then immediately frozen. A total of 400 μL of saliva was used for the RNA extraction according to the manufacturer’s recommendations, optionally with DNase I treatment. Next, cDNA syntheses were conducted using an ExScript RT reagent kit (TaKaRa). Quantitative polymerase chain reaction (qPCR) was performed using the following primer sets: human interleukin 1 beta (*IL1β*) [25] forward, 5′-ATGATGGCTTATTACAGTGGCAA-3′; human *IL1β* reverse, 5′-TGTGATGCGGT-TTAGCTGAG-3′; human tumor necrosis factor (*TNFα*) [26] forward, 5′-CCTCTCTCTAATCAGCCCTCTG-3′; human *TNFα* reverse, 5′-GAGGACCTGGGAGTAGATGAG-3′; human glyceraldehyde-3-phosphate dehydrogenase (*GAPDH*) forward, 5′-GCACCGTCAAGGCTGAGAAC-3′; human *GAPDH* reverse, 5′-ATGGTGGTGAAGACGCCAGT-3′; and SYBR Premix Ex Taw in a TP800 Thermal Cycler Dice Real-Time System (TaKaRa) [27]. The levels of *TNFα* and *IL1β* mRNA relative to the level of *GAPDH* were analyzed and used for the evaluation of oral inflammation and correlations with environmental and nutritional factors.

### 2.6. Statistical Analysis

We analyzed the environmental factors (e.g., age, obesity, and smoking) of chronic periodontitis as well as nutritional factors to clarify whether they reflected the parameters of chronic periodontitis.

The Wilcoxon signed-rank test was used to evaluate differences in clinical parameters between BL, RE, SPT1, and SPT2. Spearman’s rank correlation (*r_s_*) was used to analyze the correlations between clinical parameters and environmental and nutritional factors.

To clarify the causal dependencies between clinical parameters and environmental and nutritional factors that showed a relatively high correlation (|*r*_s_| > 0.15), multiple linear regression analysis was conducted, deriving the standardized partial regression coefficient, adjusted multiple correlation coefficient (R), adjusted coefficients of determination (R2), and regression variations. Objective variables were set to eight clinical parameters, and explanatory variables were set to four environmental and ten nutritional factors. No linear combinations between these explanatory variables were indicated based on a variance inflation factor (VIF) < 10. To determine whether there was an autocorrelation between error terms (difference between the measured and theoretical values), it was confirmed that the Durbin–Watson ratio took values between −2.0 and 2.0. Furthermore, mean absolute errors (MAEs) were calculated to evaluate the accuracy of each multiple linear regression analysis. Residual plots were used to evaluate regularity and distribution visually, and heteroskedasticity was statistically evaluated by the Breusch–Pagan test and the White test.

Stratified descriptive statistics were calculated between clinical parameters and environmental and nutritional factors based on the results of the multiple regression analysis. Five clinical parameters (number of teeth, PD, rate of PD of 4–5 mm, PISA, and PESA) were stratified by the clinical standard [14,28] and used for the analysis of explanatory variables from environmental factors. For explanatory variables in nutritional factors, PD, rate of PD of 4–5 mm, the BOP rate, PISA, and PISA/PESA were selected. Then, the correlation ratio (η^2^) and differences in the population mean were calculated and evaluated by t-tests or Welch’s t-tests according to statistical normality and equal variance. A *p* value < 0.05 was considered to indicate statistical significance. Spearman’s rank correlation (*r_s_*) was used to analyze the correlations between TNFα mRNA levels, IL1β mRNA levels, environmental factors, nutritional factors, and clinical parameters.

## 3. Results

### 3.1. Profiling for the Periodontal Examinations and Questionnaire

The periodontal examinations, saliva collection, and lifestyle questionnaire were conducted on 112 patients. A total of six patients were excluded because of incomplete periodontal examinations over four stages and insufficient salivary RNA concentrations to analyze qPCR. Finally, 106 patients were included in the analysis. The patients’ profiles and medians of the periodontal examinations were shown in each treatment stage. Most clinical parameters were improved at SPT1 and maintained at SPT2, although the BOP rate showed a tendency to increase between SPT1 and SPT2 (Appendix A). These results suggested that the study participants had received appropriate periodontal therapy. The results of the questionnaires for environmental factors are shown by a histogram (Appendix A) and a frequency table (Appendix A). For nutritional factors, the results are demonstrated by a histogram (Appendix A) and a frequency table (Appendix A). The results for the environmental factors obtained from the questionnaires showed that 25 patients had a Brinkmann index > 200 and 28 patients had a BMI > 25 kg/m^2^. The histogram of nutritional factors demonstrated that most were close to the normal distribution. However, rice, bread, and yogurt scores were concentrated around six or seven, and eggs, soybeans, natto, dark green vegetables, and Chinese cabbage showed bimodality (Appendix A).

### 3.2. Correlation Coefficient Matrix between Clinical Parameters and Environmental and Nutritional Factors

Regarding the results of the calculated correlation coefficients, the correlation coefficients were demonstrated for eight items of clinical parameters, four items of environmental factors, and 10 items of nutritional factors (at least two items with |*r_s_*| > 0.15) (Figure 1). Among the environmental factors, weak negative correlations were found between the number of teeth and age, smoking status and the Br index, and PESA and the Br index. In contrast, weak positive correlations were found between PD, smoking status, and BMI and between the rate of PD of 4–5 mm, smoking status, and BMI. No correlations were found between the BOP rate, PISA/PESA, and four environmental factors. Regarding nutritional factors, weak negative correlations were found between PD and bread, nonfatty fish, and yogurt; between the rate of PD of 4–5 mm and pork, beef, mutton, nonfatty fish, and yogurt; between the rate of PD ≥ 6 mm and pork, beef, mutton, egg, soy, tofu, dark green vegetables, and mushrooms; between the BOP rate and milk and mushrooms; between PISA and bread and dark green vegetables; and between PISA/PESA and bread, soy, milk, and mushrooms. No positive correlations were found among eight clinical parameters and ten environmental factors (Figure 1). These results suggested correlations between the loss of multiple nutritional factors and periodontal status, secondary to those between tooth loss, aging, and smoking status.

### 3.3. Multiple Linear Regression Analysis

Next, eight clinical parameters with high correlations were used as response variables, and four items in the environmental factors and ten nutritional factors were used as explanatory variables for multiple linear regression (Figure 1). The determination coefficient was calculated, and multiple regression analysis was conducted to confirm whether there were correlations between clinical parameters and environmental and nutritional factors (Table 1 and Table 2). The results showed that the clinical parameters with a determination coefficient exceeding 0.2 were the number of remaining teeth, and exceeding 0.1 was the rate of PD of 4–5 mm as a related environmental factors. These findings demonstrated a statistically significant relation to age, the Br index, and BMI as explanatory variables (Table 1). Clinical parameters with negative correlations and coefficients greater than 0.1 with nutritional factors were PD, rate of PD of 4–5 mm, BOP rate, PISA and PISA/PESA (Table 2). The MAEs of the clinical parameters were calculated by subtracting the predicted from actual values. The MAEs of most nutritional factors tended to be lower than those of the environmental factors. In the results of the residual plot using the Breusch–Pagan test and White’s test for heteroskedasticity in the multiple regression analysis, “doubtful” plots that were outliers were seen for the rates of PD of 4–5 mm and PD ≥ 6 mm as explanatory variables. However, these values were not excluded from the analysis because the errors of clinical examinations were not clarified (Appendix A).

These results suggested that environmental factors reflected long-term pathophysiology, such as the number of teeth, in clinical parameters. On the other hand, nutritional factors reflected inflammatory activity, such as the BOP rate, PISA, and PISA/PESA. Moreover, these were determined by bread, milk, and dark green vegetables, suggesting that the clinical parameters were influenced by malnutrition.

### 3.4. Stratified Descriptive Statistics between Clinical Parameters and Environmental and Nutritional Factors

Based on the results of the multiple regression analysis, the environmental and nutritional factors were selected for stratified descriptive statistics between the clinical parameters and environmental and nutritional factors. Each clinical parameter was stratified into two groups by the clinical standard. The results showed that patients with <20 teeth had a significantly larger Br index. In addition, patients with a PD ≥3.0 had a significantly larger BMI and consumed less yogurt. Patients with a rate of PD of 4–5 mm and a BOP rate ≥2.0 were significantly more likely to be current smokers. A BOP rate ≥2.0 was significantly related to lower consumption of dark green vegetables. Patients with a PISA ≥232 were older and consumed significantly less bread and dark green vegetables. Patients with a PISA/PESA ≥0.22 consumed less bread (Table 3 and Table 4). These results are represented by box plots to visualize the variations (Figure 2).

### 3.5. Stratified Descriptive Statistics between Clinical Parameters and Environmental and Nutritional Factors

Correlation coefficients between IL1β and TNFα mRNA levels in saliva and environmental and nutritional factors were evaluated. The results showed weak positive correlations between IL1β mRNA levels and smoking status but no correlation between TNFα mRNA levels and environmental factors. A very weak negative correlation was seen between IL1β mRNA levels and dark green vegetables but not between TNFα mRNA levels and nutritional factors. Interestingly, IL1β mRNA levels were weakly correlated to the BOP rate, PISA and PISA/PESA, reflecting inflammatory activity, whereas no correlation was found between TNFα mRNA levels and a rate of PD > 4–5 mm or PESA. These results suggest that IL1β mRNA levels reflected clinical parameters that imply inflammatory activities, which may be influenced by a lack of dark green vegetables (Figure 3).

## 4. Discussion

The majority of clinical parameters at SPT2 were improved compared with those at BL and RE. The patients in this study were confirmed to have received appropriate SPT (Appendix A).

The results of the histogram showing nutritional and environmental factors demonstrated three distribution patterns: close to the normal distribution, a tendency to deviate toward the highest frequency, and a bimodal distribution; however, none of the factors showed multicollinearity (data not shown).

Poor nutritional balance is well-known to weaken the resistance of periodontal tissue. Deficiencies in vitamin C, vitamin E, and carotenoids, which have antioxidant properties, weaken the immune response [29]. Additionally, low consumption of dairy products, soy products, nuts, mackerel, and salmon induces deficiencies in calcium and magnesium, which are necessary minerals for strong teeth and bones and affect the pathophysiology of periodontal disease [30]. The purpose of this research was not only to conduct an epidemiological study of the association between nutritional factors and periodontal disease but also to identify the clinical parameters of periodontal disease that reflect nutritional factors through a detailed statistical analysis of inflammatory cytokine RNA levels in saliva from patients during SPT.

Smoking status is the most influential environmental factor in periodontal disease [31]. The Br index, which is the product of the number of years of smoking and the number of cigarettes per day, is also an appropriate parameter for examining the effects of smoking over time [32]. In this study, an analysis of the relationship between these two smoking-related factors and clinical parameters revealed weak correlations with the number of teeth (*r_s_* = −0.317), PD (*r_s_* = 0.220), and the rate of PD of 4–5 mm (*r_s_* = −0.243) (Figure 1). Significant differences were also observed between the two groups classified based on the clinical standard (Table 3, Figure 2). In the results of a multiple regression analysis using the number of teeth as the response variable and age and the Br index as the explanatory variables, the coefficient of determination (R2) was 0.2304, and the standardized partial coefficients were −0.2811 and −0.2354, respectively. Although the cause of tooth defects in this study was unclear, it can be said that the effects of aging and the Br index on the number of teeth appear to be statistically almost the same. The associations between PESA, age, and Br index also showed equivalent results. However, it is difficult to conclude that PESA was purely reflected by age and the Br Index because the number of teeth affects the calculation of PESA [17] (Table 1 and Table 3).

Regarding the association between nutrition factors and clinical parameters, 10 clinical parameters showed weak correlations with at least one or more nutritional factors (Figure 1). In the results of the multiple regression analysis using these nutritional factors as explanatory variables with each clinical parameter as a response variable, the clinical parameters that showed a coefficient of determination (R2) ≥ 0.100 were PD with yogurt (R2 = 0.1256, standard partial regression coefficient = −0.2288), rate of PD of 4–5 mm with nonfatty fish and yogurt (R2 = 0.1333, standard partial regression coefficient = −0.1919, −0.2457), BOP rate with milk and dark green vegetables (R2 = 0.1256, standard partial regression coefficient = −0.2431, −0.1672), PISA with bread and dark green vegetables (R2 = 0.1204, standard partial regression coefficient = −0.1950, −0.2110), and PISA/PESA with bread and milk (R2 = 0.1384, standard partial regression coefficient = −0.2061, −0.2305) (Table 2). Moreover, the clinical parameters that showed significant differences in the comparison between the two groups were yogurt to PD, dark green vegetables to the BOP rate, bread and dark green vegetables to PISA, and bread to PISA/PESA (Table 4, Figure 2). Most R2 from nutritional factors were smaller than those from environmental factors in the multiple regression analysis. However, considering that R2 from the Br index, which evaluates the accumulated effects of smoking, was used as an explanatory variable with the number of teeth as the response variable, the magnitude of R2 from nutritional factors was reasonable (Table 2).

To analyze whether environmental and nutritional factors affect the levels of inflammatory cytokines (TNFα and IL1β) in saliva, Spearman’s rank correlation coefficients were calculated. The results showed a weak positive correlation between smoking and IL1β mRNA levels and a slight negative correlation between dark green vegetables and IL1β mRNA levels. In addition, IL1β mRNA levels correlated with clinical parameters that reflect inflammatory activity, such as the BOP rate, PISA, and PISA/PESA. These results suggest that a low intake of dark green vegetables is associated with current gingival inflammation (Figure 3).

In this study, dark green vegetables containing β-carotene were reflected in the clinical parameters of periodontal disease (*r_s_* = −0.222 with rate of PD ≥ 6 mm, *r_s_* = −0.189 with BOP rate and PISA/PESA, *r_s_* = −0.227 with PISA) (Figure 1). β-carotene is an antioxidant or anti-cancer vitamin A precursor. It has been reported that the consumption of dark green vegetables helps decrease insulin resistance by increasing β-carotene levels in the blood of patients with diabetes [33]. In patients with moderate/severe periodontitis, lower levels of β-carotene in the blood were found compared with healthy patients [34], concomitantly with low levels of the antioxidant carotenoid β-cryptoxanthin [35]. These previous reports support the finding that the BOP rate, PISA, and PISA/PESA were reflected in the frequency of intake of dark green vegetables. The changes in these indicators of inflammatory activity accompanied by nutritional status may aid the development of more effective nutritional guidance for patients with periodontal disease during SPT.

Tomatoes contain a copious amount of lycopene, which is noted for its antioxidant effects, similar to β-carotene, and has also been reported to be related to periodontal conditions. In experimental mice with periodontitis, significant improvements in periodontal healing were observed after the oral administration of lycopene compared with the control group [36]. In addition, the oral administration of lycopene has been shown to suppress IL1β levels in saliva and thereby periodontitis [37]. In clinical studies, treatment sites with lycopene for periodontitis showed a significant reduction in PD [38]. The findings of these previous reports suggest that the antioxidant effects of lycopene reduce inflammatory cytokine levels and improve the condition of periodontal tissue. In the present study, it was unclear into which of the questionnaire items tomatoes fit. However, the correlation coefficients between IL1β RNA levels in saliva and the frequency of intake of dark green vegetables (*r_s_* = −0.166), BOP (*r_s_* = 0.252), PISA (*r_s_* = 0.337), and PISA/PESA (*r_s_* = 0.279) were confirmed, likely reflected by the antioxidant effects of dark green vegetables (Figure 3).

Numerous studies have reported the effects of nutritional factors on periodontitis with bone metabolism. Fucoxanthin is a major xanthophyll and unique carotenoid with allen binding, found in brown algae, such as wakame and kelp. It is noted for its multifunctionality, such as antioxidant properties, apoptosis-inducing activity, antitumor effects, and anti-obesity effects. In models of periodontitis, fucoxanthin was shown to inhibit the promotion of osteoclast differentiation and reduce TNF, IL-1β, and IL-6 levels in blood [39]. In the present study, no association was found between seaweed intake and clinical periodontal parameters (|*r_s_*| < 0.150 with all clinical parameters) (data not shown). Astaxanthin, a carotenoid classified as a xanthophyll that is known for its anti-osteoporotic effects, is contained in red-colored natural seafood such as salmon, salmon roe, shrimp, and crab. Administering astaxanthin to periodontal sites enhanced osteoblast activity and reduced osteoclast activity, and thereby, alveolar bone resorption was attenuated in an experimental ligation-induced periodontitis model [40]. No association between milk in relation to the dynamics of Ca and clinical parameters (|*r_s_*| < 0.150 with all clinical parameters) was found in the present study (data not shown). However, correlations were observed between yogurt intake and PD (*r_s_* = −0.234), likely indicating an effect on the immune response to periodontal pathogens and bone metabolism (Figure 1). The astaxanthin-rich items were included in the low-fat fish category, with squid, octopus, shrimp, crab, and salmon in the questionnaire of the present study. However, the relationship between clinical parameters and shrimp could not be accurately obtained because a single item of shrimp did not exist in the questionnaire. It will be necessary to address this issue using a questionnaire with a single item in a future study.

All-trans retinoic acid (ATRA) is another vitamin A derivative. The administration of ATRA inhibited inflammatory cell infiltration into periodontal tissue and alveolar bone resorption in a *Porphyromonas gingivalis*-derived lipopolysaccharide-induced experimental periodontitis model [41]. It was also reported that ATRA suppressed the destruction of tight junctions of gingival epithelial cells by *P. gingivalis* in an in vitro study [11]. However, no correlation coefficient was found between carrot and squash intake and clinical periodontal parameters (|*r_s_*| < 0.150 with all clinical parameters) in this study (data not shown).

The intake of yogurt was correlated with PD (*r_s_* = −0.234), rate of PD of 4–5 mm (*r_s_* = −0.257), BOP rate (*r_s_* = −0.198), and PISA/PESA (*r_s_* = −0.163) (Figure 1). It is well-known that yogurt contains rich *Lactobacillus* spp., which helps prevent periodontitis by reducing periodontal pathogens. *Lactobacillus helveticus* SBT2171 (LH2171) has been shown to upregulate β-defensin production in oral epithelial cells in vitro and reduce *P. gingivalis*-induced proinflammatory cytokine expression in gingival epithelial cells [42]. The oral administration of *Lactobacillus gasseri* SBT2055 (LG2055) in *P. gingivalis*-infected mice was found to reduce inflammatory cytokines, thereby inhibiting the destruction of periodontal tissue [43]. These previous reports clearly indicate that the oral administration of *Lactobacillus* spp. indirectly prevents periodontitis in relation to the gut immune system.

## 5. Conclusions

In conclusion, the findings of this study indicate that the frequency and intake of dark green vegetables and yogurt, which are nutritional factors for periodontal disease, can be effective endpoints for assessing the inflammatory activity of periodontal tissue by BOP, PISA, and PISA/PESA.

## Figures and Tables

**Figure 1 nutrients-14-04993-f001:**
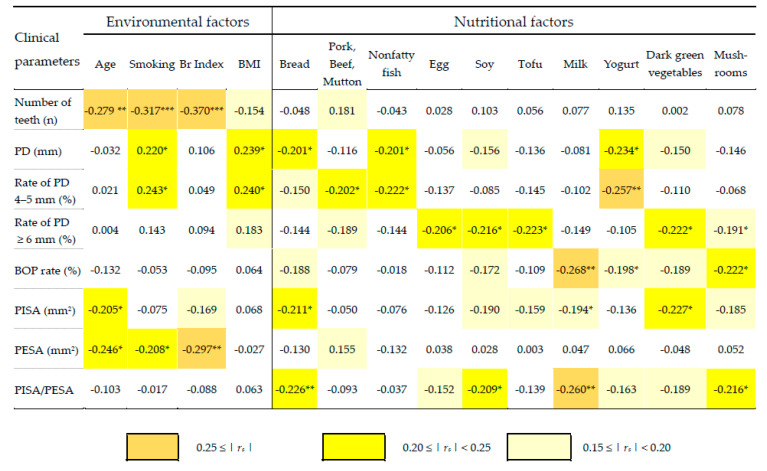
Correlation coefficient matrix. Spearman’s rank correlation coefficient (*r_s_*) between eight periodontal parameters, four environmental factors, and ten nutritional factors at SPT2. Boxes in the matrix are painted by color in response to correlation coefficient as shown above. Student’s t-distribution was used for comparisons of differences in correlation coefficients (* *p* < 0.05, ** *p* < 0.01, *** *p* < 0.001). Abbreviations: BMI, body mass index; Br index, Brinkman index; PD, probing depth; BOP, bleeding on probing; PISA, periodontal inflamed surface area; PESA, periodontal epithelial surface area.

**Figure 2 nutrients-14-04993-f002:**
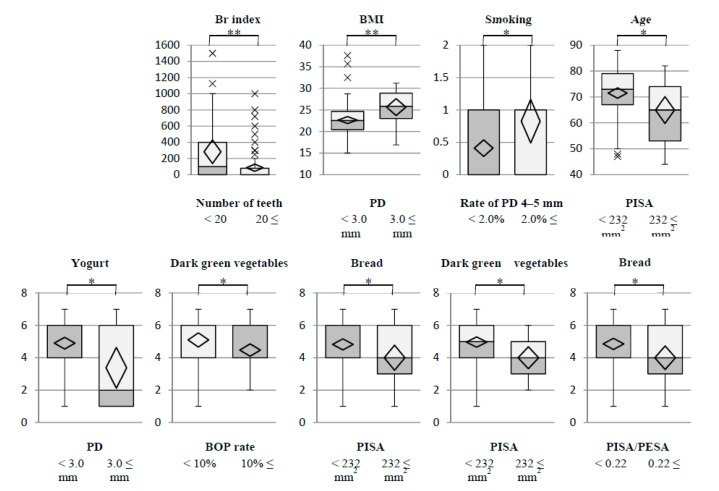
Box plot of environmental and nutritional factors. Differences in the population mean from clinical parameters are shown. The environmental and nutritional factors were limited based on statistically significant differences in the results of stratified descriptive statistics (Table 3 and Table 4). Abbreviations: Br index, Brinkman index; BMI, body mass index; PD, probing depth; BOP, bleeding on probing; PISA, periodontal inflamed surface area; PESA, periodontal epithelial surface area. t-tests or Welch’s t-tests were used to compare differences in the population mean (* *p* < 0.05, ** *p* < 0.01).

**Figure 3 nutrients-14-04993-f003:**
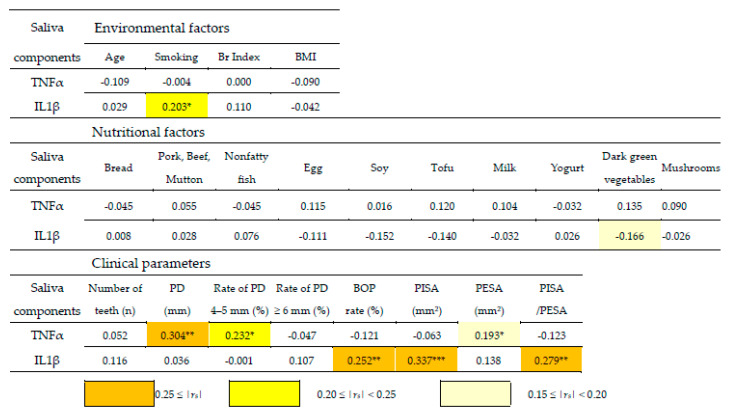
Correlation coefficients between inflammatory cytokines, environmental and nutritional factors, and clinical parameters. Spearman’s rank correlation coefficient (*r_s_*) between four environmental factors, ten nutritional factors, eight periodontal parameters, and two saliva inflammatory cytokines. Boxes in the matrix are colored based on the correlation coefficients as shown above. Student’s t-distribution was used to compare differences among correlation coefficients (* *p* < 0.05, ** *p* < 0.01). Abbreviations: TNFα, tumor necrosis factor α; IL1β, interleukin 1β; Br index, Brinkman index; BMI, body mass index; PD, probing depth; BOP, bleeding on probing; PISA, periodontal inflamed surface area; PESA, periodontal epithelial surface area.

**Table 1 nutrients-14-04993-t001:** Multiple linear regression analysis to predict clinical parameters based on environmental factors.

	Environmental Factors							
		Multiple Regression Analysis ^1^						
Clinical Parameters	Correlation Coefficient |*r_s_*| > 0.15	* *p* < 0.05 ** *p* < 0.01	Standardized Partial Regression Coefficient	Collinearity Statistics		R R^2^ Regression Variation	DW	MAE
				Tolerance	VIF			
Number of teeth	•Age •Smoking •Br index	•Age ** •Smoking •Br index * •Constant term **	−0.2811 −0.2045 −0.2354 -	0.9764 0.6567 0.6586	1.0241 1.5228 1.5185	0.4800 0.2304 *p* < 0.001	2.035	3.9
PD (mm)	•Smoking •BMI	•Smoking * •BMI * •Constant term **	-0.1960 -0.2177 -	0.9879 0.9879	1.0123 1.0123	0.3085 0.0952 *p* < 0.01	1.243	0.3
Rate of PD of 4–5 mm (%)	•Smoking •BMI	•Smoking * •BMI * •Constant term **	0.2196 -0.2156 -	0.9879 0.9879	1.0123 1.0123	0.3242 0.1051 *p* < 0.01	1.360	3.2
Rate of PD ≥6 mm (%)	•BMI	•BMI •Constant term	0.1835 -	1.0000	1.0000	0.1835 0.0337 *p* = 0.06	1.644	1.5
Rate of BOP (%)	N/A	N/A	-	-	-	-	-	-
PISA (mm^2^)	•Age •Br index	•Age * •Br index •Constant term **	−0.1959 −0.1576 -	0.9963 0.9963	1.0037 1.0037	0.2588 0.0670 *p* < 0.05	1.907	91.2
PESA (mm^2^)	•Age •Smoking •Br index	•Age * •Br index ** •Constant term **	−0.2290 −0.2831 -	0.9963 0.9963	1.0037 1.0037	0.3748 0.1450 *p* < 0.001	2.281	237.1
PISA/PESA	N/A	N/A	-	-	-	-	-	-

^1^. Multiple linear regression analysis was performed by calculating the predicted response variables (periodontal clinical parameters) based on explanatory variables for environmental factors. The assumption was satisfied, as the collinearity statistics showed a tolerance higher than 0.5 and a VIF < 2. Statistically significant differences (* *p* < 0.05, ** *p* < 0.01). Abbreviations: PD, probing depth; BOP, bleeding on probing; PISA, periodontal inflamed surface area; PESA, periodontal epithelial surface area; PCR, plaque control record; BMI, body mass index; VIF, variance inflation factor; R, adjusted multiple correlation coefficient; R2, adjusted coefficient of determination; DW, Durbin–Watson; MAE, mean absolute error.

**Table 2 nutrients-14-04993-t002:** Multiple linear regression analysis to predict clinical parameters based on nutritional factors.

	Environmental Factors							
		Multiple Regression Analysis ^1^						
Clinical Parameters	Correlation Coefficient |*r_s_*| > 0.15	* *p* < 0.05 ** *p* < 0.01	Standardized Partial Regression Coefficient	Collinearity Statistics		R R^2^ Regression Variation	DW	MAE
				Tolerance	VIF			
Number of teeth	•Pork, beef, mutton	•Pork, beef, mutton •Constant term **	0.1806 -	1.0000	1.0000	0.1806 0.0326 *p* = 0.06	1.976	4.6
PD (mm)	•Bread soy •Nonfatty fish •Yogurt •Dark green vegetables	•Bread •Nonfatty fish * •Yogurt * •Constant term **	−0.1740 −0.1847 −0.2288 -	0.9889 0.9902 0.9986	1.0113 1.0099 1.0014	0.3544 0.1256 *p* < 0.01	1.364	0.3
Rate of PD of 4–5 mm (%)	•Bread •Pork, beef, mutton •Nonfatty fish •Yogurt	•Pork, beef, mutton •Nonfatty fish * •Yogurt ** •Constant term **	−0.1342 −0.1919 −0.2457 -	0.9361 0.9445 0.9904	1.0683 1.0588 1.0097	0.3647 0.1330 *p* < 0.01	1.453	3.4
Rate of PD ≥6 mm (%)	•Pork, beef, mutton •Egg soy •Tofu •Dark green vegetables •Mushrooms	•Tofu •Dark green vegetables •Constant term **	−0.1620 −0.1600 -	0.8528 0.8528	1.1726 1.1726	0.2678 0.0717 *p* < 0.05	1.782	1.5
BOP rate (%)	•Bread soy •Milk yogurt •Dark green vegetables •Mushrooms	•Bread •Milk ** •Dark green vegetables * •Constant term **	−0.1670 −0.2413 −0.1672 -	0.9939 0.9877 0.9935	1.0061 1.0124 1.0066	0.3572 0.1276 *p* < 0.05	1.627	6.5
PISA (mm^2^)	•Bread •Soy tofu •Milk •Dark green vegetables •Mushrooms	•Bread * •Milk •Dark green vegetables * •Constant term **	−0.1950 −0.1620 −0.2110 -	0.9939 0.9877 0.9935	1.0061 1.0124 1.0066	0.3470 0.1204 *p* < 0.01	2.020	87.9
PESA (mm^2^)	•Pork, beef, mutton	•Pork, beef, mutton •Constant term **	0.1550 -	1.0000	1.0000	0.1550 0.0240 *p* = 0.11	2.141	255.4
PISA/PESA	•Bread •Egg soy •Milk yogurt •Dark green vegetables •Mushrooms	•Bread * •Milk * •Dark green vegetables •Constant term **	−0.2061 −0.2305 −0.0061 -	0.9939 0.9877 0.9935	1.0061 1.0124 1.0066	0.3720 0.1384 *p* < 0.01	1.701	0.074

^1^. In brief, all methods and representations are the same as those in Table 1. The assumption was satisfied, as the collinearity statistics showed a tolerance higher than 0.5 and a VIF < 2. Statistically significant differences (* *p* < 0.05, ** *p* < 0.01). Abbreviations: PD, probing depth; BOP, bleeding on probing; PISA, periodontal inflamed surface area; PESA, periodontal epithelial surface area; PCR, plaque control record; BMI, body mass index; VIF, variance inflation factor; R, adjusted multiple correlation coefficient; R2, adjusted coefficient of determination; DW, Durbin–Watson; MAE, mean absolute error.

**Table 3 nutrients-14-04993-t003:** Stratified descriptive statistics between clinical parameters and environmental factors.

	Environmental Factors							
Clinical Parameters	Multiple Regression Analysis Explanatory Variables	Stratified Descriptive Statistics				Testing of Differences of Population Mean		
		Stratified Standard of Response Variables	Numbers	Mean ± SD of Explanatory Variables	Correlation Ratio (η^2^) * *p* < 0.05 ** *p* < 0.01	Hypothesis Testing for the Homogeneity of Variances	Methods	*p* Value Statistical Power
Number of teeth	Age **	<20 20≤	31 75	72.7 ± 7.52 69.2 ± 10.83	0.0249	*p* < 0.05	*t*-test	*p* = 0.10 0.3650
	Br index *			282.71 ± 396.86 85.800 ± 194.24	0.1016 **	*p* < 0.001	*t*-test	*p* < 0.001 ** 0.9248
PD (mm)	Smoking *	<3.0 3.0≤	16 90	0.456 ± 0.621 0.750 ± 0.775	0.0265	*p* = 0.21	Welch’s *t*-test	*p* = 0.17 0.2769
	BMI *			22.7 ± 3.81 25.7 ± 3.68	0.0759 **	*p* = 0.94	Welch’s *t*-test	*p* < 0.01 * 0.8150
Rate of PD of 4–5 mm (%)	Smoking *	<2.0 2.0≤	41 65	0.369 ± 0.575 0.707 ± 0.716	0.0646 **	*p* = 0.12	Welch’s *t*-test	*p* < 0.05 * 0.7109
	BMI *			22.8 ± 4.016 23.6 ± 3.767	0.0091	*p* = 0.67	Welch’s *t*-test	*p* = 0.32 0.1660
PISA (mm^2^)	Age *	<232 232≤	85 21	71.5 ± 9.35 65.0 ± 11.4	0.0681 **	*p* = 0.22	Welch’s *t*-test	*p* < 0.05 * 0.6550
	Br index			153.99 ± 299.1 100.48 ± 200.8	0.0058	*p* < 0.05	*t*-test	*p* = 0.33 0.1606
PESA (mm^2^)	Age *	<1026 1026≤	75 31	71.2 ± 9.77 67.9 ± 10.6	0.0220	*p* = 0.58	Welch’s *t*-test	*p* = 0.14 0.3065
	Br index **			163.52 ± 301.8 94.68 ± 225.40	0.0124	*p* = 0.08	Welch’s *t*-test	*p* = 0.20 0.2464

Based on the results of multiple regression analysis, environmental factors were selected, and subjects were stratified by clinical standard. The correlation ratio (η^2^) and differences in the population mean were calculated. Abbreviations: PD, probing depth; PISA, periodontal inflamed surface area; PESA, periodontal epithelial surface area; BOP, bleeding on probing.

**Table 4 nutrients-14-04993-t004:** Stratified descriptive statistics between clinical parameters and environmental factors.

	Nutritional Factors							
Clinical parameters	Multiple Regression Analysis Explanatory Variables	Stratified Descriptive Statistics				Testing of Differences of Population Mean		
		Stratified Standard of Response Variables	Numbers	Mean ± SD of Explanatory Variables	Correlation Ratio (η^2^) * *p* < 0.05 ** *p* < 0.01	Hypothesis Testing for the Homogeneity of Variances	Methods	*p* Value Statistical Power
PD (mm)	Nonfatty fish *	<3.0 3.0≤	31 75	3.26 ± 1.11 2.69 ± 1.14	0.0330	*p* = 0.81	Welch’s *t*-test	*p* = 0.08 0.4207
	Yogurt *			4.90 ± 1.74 3.38 ± 2.36	0.0824 **	*p* = 0.09	Welch’s *t*-test	*p* < 0.05 * 0.6454
Rate of PD of 4–5 mm (%)	Nonfatty fish *	<2.0 2.0≤	16 90	3.19 ± 1.07 3.15 ± 1.22	0.0003	*p* = 0.37	Welch’s *t*-test	*p* = 0.87 0.0530
	Yogurt *			4.94 ± 1.66 4.24 ± 2.21	0.0316	*p* = 0.94	*t*-test	*p* = 0.07 0.4468
BOP rate (%)	Milk **	<10.0 10.0≤	51 55	4.67 ± 2.02 4.01 ±2.09	0.0211	*p* = 0.81	Welch’s *t*-test	*p* = 0.13 0.3309
	Dark green vegetables *			5.10 ± 1.57 4.47 ± 1.41	0.0430 *	*p* = 0.46	Welch’s *t*-test	*p* < 0.05 * 0.5687
PISA (mm^2^)	Bread *	<232 232≤	85 21	4.82 ± 1.59 4.00 ± 1.70	0.0405 *	*p* = 0.64	Welch’s *t*-test	*p* < 0.05 * 0.5465
	Dark green vegetables *			4.97 ± 1.47 4.00 ± 1.48	0.0651 **	*p* = 0.89	Welch’s *t*-test	*p* < 0.05 * 0.7355
PISA/PESA	Bread *	<0.22 0.22≤	82 24	4.85 ± 1.57 4.00 ± 1.72	0.0480 *	*p* = 0.55	Welch’s *t*-test	*p* < 0.05 * 0.5636
	Milk *			4.45 ± 2.03 4.00 ± 2.21	0.0084	*p* = 0.56	Welch’s *t*-test	*p* = 0.38 0.1408

In brief, all methods and representations are the same as those in Table 3. Abbreviations: PD, probing depth; PISA, periodontal inflamed surface area; PESA, periodontal epithelial surface area; BOP, bleeding on probing.

## Data Availability

The data cannot be made available due to privacy restrictions.

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
