# Peer review of "Association between Dietary Habit and Clinical Parameters in Patients with Chronic Periodontitis Undergoing Supportive Periodontal Therapy"

_nutrients, 2022, doi:10.3390/nu14234993_

Round 1
Reviewer 1 Report
Thank you for the opportunity to read this interesting article. In the introduction, the authors should start from the quality of life, and in particular from the quality of chewing related to the state of health. They should supplement with specific and general purpose questionnaires that can be used to assess oral diseases and supplement the literature with El Osta N, Pichot H, Soulier-Peigue D, Hennequin M, Tubert-Jeannin S. Validation of the child oral health impact profile (COHIP) french questionnaire among 12 years-old children in New Caledonia. Health Qual Life Outcomes. 2015 Oct 30;13:176. doi: 10.1186/s12955-015-0371-9. PMID: 26518886; PMCID: PMC4628352.; Zhu H, Zhou H, Qin Q, Zhang W. Association between Smoking and Sugar-Sweetened Beverage Consumption, Tooth Brushing among Adolescents in China. Children (Basel). 2022 Jul 6;9(7):1008. doi: 10.3390/children9071008. PMID: 35883992; PMCID: PMC9319217.: Chmielik LP, Mielnik-Niedzielska G, Kasprzyk A, Stankiewicz T, Niedzielski A. Health-Related Quality of Life Assessed in Children with Chronic Rhinitis and Sinusitis. Children (Basel). 2021 Dec 4;8(12):1133. doi: 10.3390/children8121133. PMID: 34943329; PMCID: PMC8699909
Did the exclusion criteria include acute oral cavity inflammations and abscesses around the teeth / please add these units to the criteria for inclusion in the study group or exclusion
Author Response
I would like to thank you and reviewers for reviewing our manuscript. All the comments made by the reviewers were truly constructive and contributed to the further improvement of our revised manuscript. We highlighted the changes within the document by using yellow color at the pointed out by reviewers. We believe it is better that reviewers and readers can understand this work and have more interests in periodontal therapy considering nutritional factors.Our responses to their comments are listed below. We extensively revised our manuscript accordingly.

Reviewer 2 Report
This is a good manuscript on technically correct and sums up exciting results.
This manuscript complements the literature with data on the correlations between dietary habits and clinical results of patients with chronic periodontitis under supportive periodontal therapy.
The manuscript is well-written and presents useful methods for the study of the aspects analyzed, and sums up interesting results.
This article highlights that environmental and nutritional factors are useful for assessing the risk of periodontitis during supportive periodontal therapy.
I suggest some revisions:
I have complaints about the layout of some tables and their legends, they are not of good quality and don`t respect the journal requirements, please consider improving them. The same with figures.
Line 151-158, no need for the spaces.
The introduction part needs to be rewritten and improved.
Also, Lines 75-83 – this paragraph seems to be a description good for material and method, I don`t see the right place for this information in the introduction.
The discussion and conclusions are not so clear. This part can be improved with more details, explanations, and correlations of the results obtained.
Author Response

(The authors gave the same response as above.)

Round 2
Reviewer 1 Report
I accept in present form